# Case Report: Atypical Nodular Dermatofibrosis and Renal Cysts in a Bichon Frise with a BRCA2 Mutation and No FLCN Mutation

**DOI:** 10.3390/ani15142070

**Published:** 2025-07-14

**Authors:** Kwangsup Lee, Chansik Nam, Taejung Dan, Kijong Lee, Heemyung Park

**Affiliations:** 1Department of Veterinary Internal Medicine, College of Veterinary Medicine, Konkuk University, Seoul 05029, Republic of Korea; deu02181@naver.com (K.L.); enzmahsnam@naver.com (C.N.); dantae162534@naver.com (T.D.); 2Royal Animal Medical Center, Jungnang-gu, Seoul 02140, Republic of Korea; koreasvcd@gmail.com

**Keywords:** nodular dermatofibrosis, renal cysts, Bichon Frise, BRCA2 mutation

## Abstract

Nodular dermatofibrosis (ND) is a rare disease causing multiple firm skin nodules, often linked to kidney cysts or tumors. It typically affects specific breeds like the German Shepherd and is associated with mutations in the folliculin (FLCN) gene. This case describes a Bichon Frise, a breed not previously linked to the disease and lacking the usual FLCN mutation. Instead, the dog had a BRCA2 gene mutation, known for roles in DNA repair and tumor suppression. The dog showed both skin nodules and kidney cysts—a combination not previously reported in this breed. Tests confirmed the nodules were not cancerous but showed abnormal collagen accumulation. The kidney cysts’ type could not be confirmed. This case suggests ND may occur in other breeds and result from genetic changes beyond FLCN. It highlights the need to explore diverse genetic factors in the disease’s development. Greater understanding can help veterinarians diagnose and manage the condition more effectively, improving the welfare of affected dogs.

## 1. Introduction

ND is a rare hereditary disorder commonly reported in German Shepherd dogs [1,2,3,4,5,6,7]. It is characterized by the development of multiple firm cutaneous nodules, often accompanied by renal lesions such as cystadenomas or cystadenocarcinomas. In some cases, nodular lesions have also been observed in organs other than the kidneys, including the uterus, ovaries, myocardium, and lungs [7]. This condition is widely believed to be associated with mutations in the FLCN gene [5,8], which functions as a tumor suppressor and plays a critical role in regulating cell growth and collagen synthesis. Loss of FLCN function is thought to contribute to excessive collagen accumulation in the skin and epithelial proliferation in the kidneys.

Although initially described predominantly in German Shepherds, ND has since been observed in various other breeds, including German Shorthaired Pointers [9], Boxers [10], Golden Retrievers [11,12], Beagles [13], Australian Cattle Dogs [14], and mixed-breed dogs [8,10,15,16]. Furthermore, recent studies have reported ND-like clinical presentations even in the absence of detectable FLCN mutations [12]. Additionally, some affected dogs have presented with only cutaneous lesions without concurrent renal or uterine abnormalities [9,12,14]. These findings suggest that ND may occur independently of FLCN mutations and may not always represent a paraneoplastic manifestation of renal tumors.

In this report, a case is described of concurrent ND and renal cystic lesions in a Bichon Frise without any identifiable FLCN gene mutation. Notably, a mutation in the BRCA2 gene was identified. BRCA2 is a tumor suppressor gene involved in DNA repair, and its mutation raises the possibility of an alternative genetic pathway contributing to the development of ND. This case offers a novel perspective that expands the current understanding of the disease, which has traditionally focused on FLCN-related pathogenesis.

To the best of current knowledge, this is the first report describing coexisting ND and renal cystic lesions in a Bichon Frise with a concurrent BRCA2 mutation. This finding highlights the potential for broader breed susceptibility and underscores the need for further research into the genetic heterogeneity and pathophysiology of ND.

## 2. Case Description

A 10-year-old intact female Bichon Frise was presented with multiple alopecic cutaneous nodules accompanied by localized swelling (Figure 1). The patient had no history of systemic disease or previous surgical procedures. According to the owner, the nodules initially developed on the head and proximal regions of both forelimbs (humeral areas), and gradually spread to the radius and ulna. Approximately seven months later, additional nodules appeared on both hindlimbs. The dog remained clinically stable throughout the disease course, with no changes in appetite, activity, or behavior. Physical examination revealed no abnormalities other than the cutaneous nodules, and no signs of pain or discomfort were observed.

Fine-needle aspiration (FNA) was performed on multiple cutaneous nodules (Figure 2). The smears, stained with Diff-Quik, were moderately cellular and predominantly composed of spindle-shaped cells, suggestive of fibroblasts or myofibroblasts. The cells exhibited mild to moderate anisocytosis and anisokaryosis, with increased cytoplasmic basophilia. The background contained abundant eosinophilic, amorphous extracellular material, presumed to be composed of collagen, proteoglycans, glycosaminoglycans (GAGs), and elastin. A mild inflammatory infiltrate was also noted. The cytologic findings were consistent with a mature to mildly atypical mesenchymal population. However, the distinction between reactive fibroblastic proliferation and low-grade neoplasia could not be definitively made based on cytology alone.

Given the limitations of cytologic evaluation in fully characterizing the lesions, additional diagnostic tests were performed to determine the extent of lesion involvement, assess for possible metastasis, and evaluate whether the condition represented a neoplastic process or was secondary to an underlying systemic disease. Whole-body radiographic examination revealed increased soft tissue opacity throughout the subcutaneous regions of all four limbs. Focal areas of increased radiodensity and soft tissue thickening were noted. No evidence of bone invasion or definitive metastatic lesions was identified. A complete blood count and serum biochemistry analysis were performed, but no significant abnormalities were identified.

As the initial diagnostic workup, including cytology, blood tests, and imaging, failed to clarify the nature of the disease, incisional biopsies were planned to obtain a definitive diagnosis. Under local infiltration anesthesia, punch biopsies were collected from four representative sites. Each site involved a small skin incision followed by primary closure with sutures. Postoperatively, to prevent secondary infection and manage pain, a 7-day course of oral antimicrobial therapy was prescribed using cephalexin (Palexin^®^, 30 mg/kg, PO, BID; Dong Wha Pharm Co., Ltd., Seoul, Republic of Korea) and enrofloxacin (Baytril^®^, 2.5 mg/kg, PO, BID; Elanco, Greenfield, IN, USA). Analgesia was provided using tramadol hydrochloride (Tridol^®^, 3 mg/kg, PO, BID; Yuhan Corporation, Seoul, Republic of Korea). The collected biopsy specimens were submitted to Antech Diagnostics (17672 Cowan Avenue, Irvine, CA 92614, USA) for histopathologic evaluation.

Histopathologic evaluation of the biopsied tissues (Figure 3) revealed well-demarcated nodules composed of mature, hypocellular collagen. Fibrocytes and fibroblasts within the lesions appeared morphologically bland, with no evidence of pleomorphism or mitotic activity. Masson’s trichrome staining highlighted collagen fibers in blue, and both the density and staining intensity of collagen increased in proportion to the degree of dermal fibrosis. The lesions extended into the deep dermis, and in some areas, the fibrotic tissue replaced or compressed the pannicular adipose tissue, resulting in attenuation of the subcutaneous fat layer. No histologic features of malignancy were identified. Taken together, these findings were considered consistent with ND.

ND has been reported in association with renal cystadenomas or cystadenocarcinomas, and nodular lesions may also develop in other internal organs. To assess potential visceral involvement, an abdominal ultrasonographic examination was performed (Figure 4). Ultrasound evaluation revealed multiple round, smoothly marginated, anechoic cystic structures confined to the renal parenchyma of both kidneys. Most cysts exhibited distal acoustic enhancement and mildly distorted the renal contour.

Several cysts in the cranial pole of the left kidney displayed internal septations while remaining anechoic. One cyst in the mid-region of the right kidney showed mildly increased internal echogenicity but remained sharply marginated, with no vascular flow detected on color doppler imaging. No other ultrasonographic abnormalities were identified in the remaining abdominal organs.

Whole exome sequencing (WES) was performed to evaluate potential genetic mutations associated with ND. No pathogenic variants were identified in the FLCN gene, which is commonly implicated in ND and renal cystadenocarcinoma (RCND), or in other congenital hereditary disease genes. However, multiple nonsynonymous mutations were detected in the BRCA2 gene.

Prednisolone (Solondo^®^, 0.5 mg/kg, PO, SID; Yuhan Corporation, Seoul, Republic of Korea) was administered as a supportive treatment, with the aim of achieving partial regression of the cutaneous nodules and improvement of associated localized edema. However, no noticeable reduction in nodule size or associated swelling was observed.

Two months later, follow-up evaluation was conducted, including abdominal ultrasonography and serum biochemistry. Compared to the previous examination, no significant changes were noted on ultrasound. Most blood parameters remained within normal limits, but serum creatinine had increased from 0.9 mg/dL to 1.5 mg/dL (reference range: 0.5–1.5 mg/dL), and serum phosphate had risen from 2.8 mg/dL to 4.4 mg/dL (reference range: 2.4–6.4 mg/dL), suggestive of mild progression in renal function markers.

The owner declined further diagnostic or therapeutic interventions, and no subsequent follow-up visits have been recorded to date.

## 3. Discussion

This report describes a rare case of ND accompanied by bilateral renal cystic lesions in a Bichon Frise. ND has predominantly been reported in German Shepherds, and to the authors’ knowledge, this is the first documented case in a Bichon Frise. Genetic analysis revealed no mutation in the FLCN gene, which is commonly associated with ND and RCND, but multiple nonsynonymous variants were identified in the BRCA2 gene. Abdominal ultrasonography showed multiple cystic structures in both kidneys. These findings suggest that a syndrome resembling RCND may occur in Bichon Frises without FLCN mutations, and that BRCA2 may represent a potential alternative genetic contributor.

The etiology of ND has been explored through multiple perspectives since its initial recognition, with diverse hypotheses proposed over time (Table 1). In the early 1980s, the condition was recurrently documented in specific dog breeds, particularly German Shepherds, suggesting a hereditary predisposition as a primary factor [1,2,3]. These reports commonly described the simultaneous occurrence of cutaneous nodules and renal cystic lesions. However, a study reported in 1985 described a case involving only cutaneous lesions and attributed the pathogenesis to localized collagen hyperplasia or hamartomatous changes, thereby suggesting a potential phenotypic variation [9].

In the 1990s, the concept of paraneoplastic syndrome was introduced [11], proposing that growth factors secreted by renal tumors could stimulate dermal fibrotic changes. Subsequent studies, however, demonstrated local overexpression of transforming growth factor beta 1 (TGF-β1) within the affected dermal tissues, implying that the pathogenesis of RCND may involve not only systemic effects but also intrinsic fibrotic processes within the skin itself [6].

In 2003, the identification of a pathogenic mutation (H255R) in the FLCN gene (also known as Birt–Hogg–Dubé syndrome gene, or BHD) provided strong evidence that the disease may have a hereditary basis [5]. Subsequent studies proposed that haploinsufficiency of FLCN leads to the initial development of cutaneous lesions, whereas renal pathology may occur later through additional somatic mutations, consistent with the “second hit” hypothesis [14].

Importantly, recent reports have documented cases of RCND in which no pathogenic mutations in the FLCN gene were detected, suggesting that the pathogenesis of the disease may not be solely attributable to FLCN alterations [12]. Furthermore, histopathological analyses of RCND lesions have revealed the activation of CD3^+^ T lymphocytes and CD163^+^ macrophages, raising the possibility that RCND represents a form of chronically active fibrosis driven by immunologic mechanisms, rather than purely hereditary fibrotic disorder [7].

In the present case, no pathogenic mutations were identified in the FLCN gene, which is classically associated with ND and RCND. However, multiple nonsynonymous variants were detected in the BRCA2 gene, suggesting that RCND-like syndromes may occur independently of FLCN alterations and that BRCA2 may represent a potential alternative genetic contributor. In the present case, the identified BRCA2 variant is presumed to be a hypomorphic mutation rather than a complete loss-of-function, raising the possibility that it contributed to the development of systemic fibrosis. Several molecular mechanisms have been proposed to explain such an association. For instance, studies have demonstrated that the TGF-β signaling pathway is activated in BRCA2-deficient cells, leading to fibroblast activation [17]. Similarly, in Fanconi anemia models harboring mutations in FA/BRCA pathway genes, including BRCA2, defective DNA repair has been shown to cause hyperactivation of the TGF-β pathway, resulting in tissue damage and fibrosis [18]. In addition, DNA damage responses triggered by genomic instability may activate the TGF-β1 pathway through the ATM–c-Cbl–TβRII axis [19].

Further supporting this link, BRCA1/2 mutations associated with homologous recombination deficiency-low (HRD-low) status have been shown to selectively activate the TGF-β pathway [20], suggesting a mechanism that could also apply to the BRCA2 variant identified in this case. Moreover, recent findings indicate that BRCA2 mutations can lead to activation of Heat Shock Factor 1 (HSF1), which in turn induces Clusterin (CLU) expression in fibroblasts. These CLU-positive fibroblasts acquire a profibrotic phenotype, contributing to excessive extracellular matrix production and tissue fibrosis [21]. This mechanism provides an additional explanation for the widespread fibrotic changes observed in the present case. Taken together, these findings suggest that the BRCA2 variant in this case may have contributed to multiorgan fibrosis through direct or indirect mechanisms involving TGF-β pathway hyperactivation and/or the induction of CLU-positive fibroblasts. Accordingly, this case offers a valuable pathophysiological insight, proposing a novel mechanistic link between BRCA2 mutations and systemic fibrotic disease.

Based on the genetic analysis of the present case in conjunction with previous research, it is evident that renal cystadenocarcinoma and ND cannot be attributed to a single etiological factor. Rather, RCND should be regarded as a heterogeneous syndrome involving multiple interacting mechanisms, including genetic abnormalities (e.g., mutations in *FLCN* or *BRCA2*), TGF-β1-mediated fibrotic signaling, immune-driven inflammation, fibrotic remodeling of the tumor microenvironment, and breed-specific genetic predispositions. This case exemplifies such multifactorial pathogenesis and may contribute to expanding and redefining the molecular understanding of RCND.

The existence of renal cystic lesions and their classification as benign or malignant have been variably discussed and reported in previous studies (Table 1). In the present case, the renal lesions observed through ultrasonography could not be clearly differentiated between cystadenoma, cystadenocarcinoma, and simple cysts, due to the lack of histological examination. However, imaging findings exclusively revealed cystic structures without any solid masses, suggesting a likelihood of non-neoplastic lesions. Previous reports have documented cases in which only simple renal cysts were present without evidence of neoplastic transformation [10], and the current case may represent a similar presentation. Several hypotheses have been proposed regarding the pathogenesis of renal cysts; among them is the theory that proliferation of epithelial cells may obstruct renal tubules, leading to localized cystic dilatation [9]. However, in the present case, histopathological confirmation of epithelial proliferation was not available, and imaging failed to reveal any mass-forming lesions. In contrast, extensive fibrosis was observed in the skin, and the kidneys exhibited only cystic structures, accompanied by rising creatinine and phosphorus levels, suggesting a decline in renal function. These findings collectively support the hypothesis that the renal cysts in this case may have developed secondarily to obstruction of the renal tubules or outflow tracts due to fibrosis, rather than as a result of epithelial proliferation [10]. Meanwhile, the possibility of polycystic kidney disease (PKD), a congenital condition, was also considered in the differential diagnosis of the renal cysts. PKD is a rare hereditary disease reported in specific breeds such as Bull Terriers [22,23], Cairn Terriers [24], and West Highland White Terriers (WHWTs) [25]. In Bull Terriers, it is known to result from a pathogenic mutation in the *PKD1* gene, and multiple renal cysts can be identified by ultrasonography. In Cairn Terriers and WHWTs, an autosomal recessive mode of inheritance has been proposed, and these breeds tend to develop cysts in both the kidneys and the liver.

However, the present case occurred in a Bichon Frise, and WES revealed no pathogenic variants in *PKD1* or in any other genes associated with congenital hereditary disorders. In addition, no hepatic lesions were observed. Taken together, these findings demonstrate distinct differences in breed, genetic background, and distribution of lesions compared to previously reported hereditary PKD cases, suggesting that the likelihood of PKD in this case is low. Nevertheless, histological evaluation is necessary to confirm the presence of such fibrotic or proliferative changes.

BRCA2 mutations are known to promote widespread fibrotic responses through various mechanisms, including activation of the TGF-β signaling pathway and induction of CLU-positive fibroblasts. It is therefore plausible that the renal cysts observed in this case are associated with BRCA2-related fibrotic processes. In addition, BRCA2 functions as a tumor suppressor gene in various tissues, and its dysfunction may increase the risk of neoplastic transformation. Although no tumorigenic lesions were identified in the present case, the possibility that the cystic lesions may progress to neoplasia over time cannot be entirely excluded. Indeed, in some RCND cases, simple cysts have been reported to undergo progression to cystadenomas or cystadenocarcinomas [2].

Although histologic confirmation of carcinoma was not obtained in this case, the findings suggest that BRCA2 mutations may contribute not only to the formation of renal cysts but also potentially to later neoplastic progression. This case thus provides valuable insight into the genetic and fibrotic mechanisms potentially underlying RCND and expands our understanding of its pathophysiology from a non-neoplastic starting point.

## 4. Conclusions

This case represents the first reported instance of RCND in a Bichon Frise, whereas previous reports have predominantly involved German Shepherds. This finding supports that RCND is not confined to a specific breed and may occur across diverse canine populations. Furthermore, the identification of a BRCA2 variant in this case suggests that, in addition to the previously recognized FLCN mutations, other genetic factors may also contribute to the development of RCND. Considering the proposed involvement of multiple pathophysiological mechanisms—such as activation of the TGF-β1 pathway, fibrotic responses, immune dysregulation, and remodeling of the tumor microenvironment—RCND may be best understood as a heterogeneous syndrome, which cannot be explained by a single etiology or clinical outcome.

Previous reports have documented cases in which simple renal cysts progressed over time to cystadenomas or cystadenocarcinomas, and some of these studies have speculated that genetic factors may be involved in such transitions. The present case, in which a BRCA2 variant was identified without concurrent evidence of neoplastic transformation, may provide supportive or complementary evidence for these hypotheses. Accordingly, long-term monitoring is warranted to evaluate potential changes in the renal lesions over time.

In summary, this case provides valuable insight into the genetic diversity and complex pathophysiology of RCND. Continued investigation into the interplay among various genetic and molecular factors is essential to better understand the clinical manifestations and prognosis of this syndrome.

## Figures and Tables

**Figure 1 animals-15-02070-f001:**
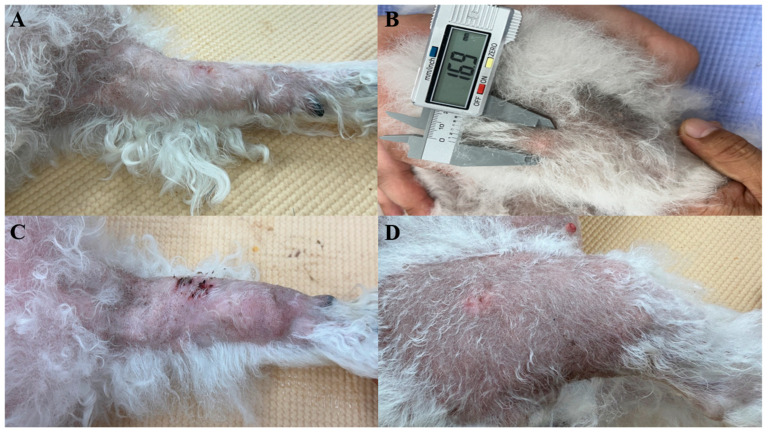
Multiple firm, mobile cutaneous nodules with surrounding alopecia were observed on all four limbs of a 10-year-old intact female Bichon Frise. The lesion on the right forelimb presented with ulceration. No pain response was elicited on palpation. (**A**) Left forelimb (9.1 mm), (**B**) left hindlimb (16.9 mm), (**C**) right forelimb (30.0 mm), (**D**) right hindlimb (8.4 mm).

**Figure 2 animals-15-02070-f002:**
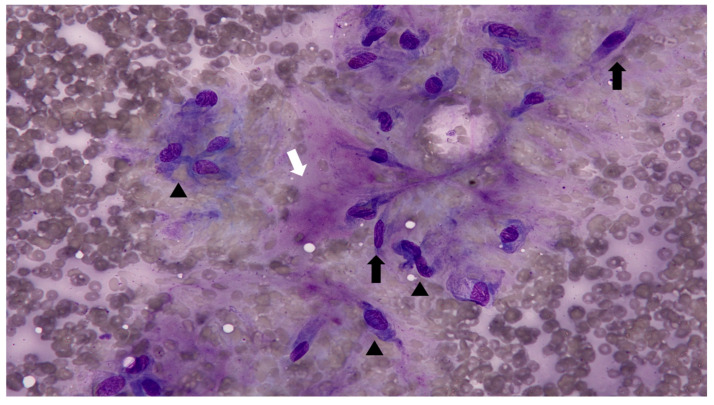
Fine-needle aspiration cytology of cutaneous nodules (Diff-Quik stain, ×400). The sample revealed spindle-shaped mesenchymal cells (black arrows) with mild to moderate anisocytosis and anisokaryosis (arrowheads) and increased cytoplasmic basophilia. Scattered eosinophilic amorphous material (white arrow), likely to represent the extracellular matrix, was present. Cellularity was low to moderate with mild cellular atypia.

**Figure 3 animals-15-02070-f003:**
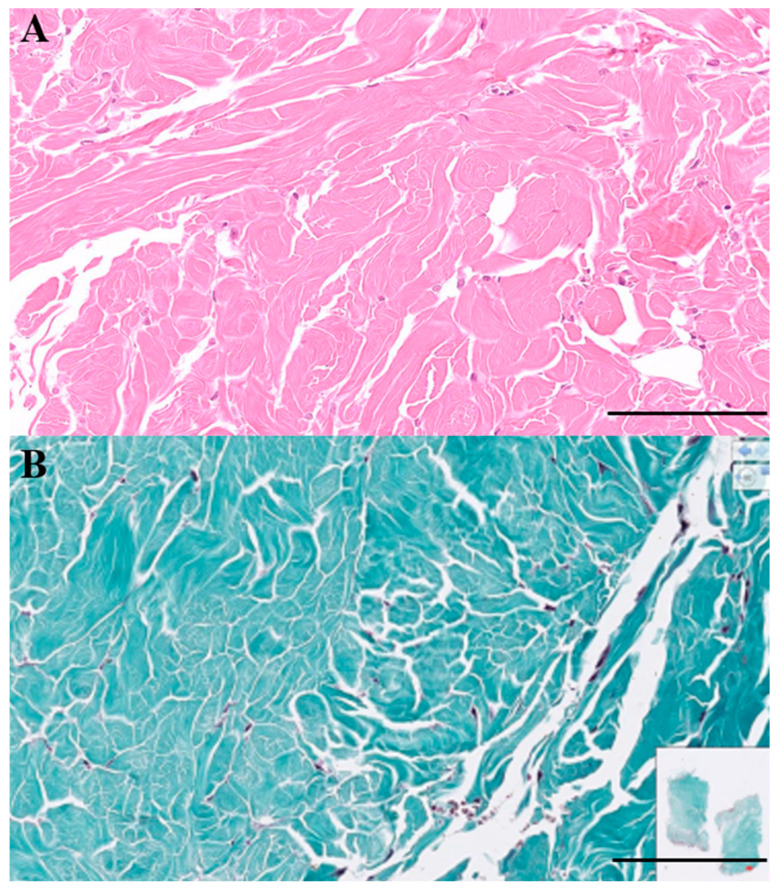
Histopathological findings of cutaneous nodules collected from the right forelimb (medial and lateral) and right hindlimb. (**A**) Hematoxylin and eosin staining revealed well-demarcated nodules composed of mature, hypocellular collagen. A thin subcutaneous fat layer separated the dermis from the nodules. Fibrocytes were not pleomorphic, and no mitotic figures were observed (×15, scale bar = 200 μm). (**B**) Masson’s trichrome staining showed increased blue staining within the interstitial and perivascular areas, confirming collagen deposition (×15, scale bar = 200 μm). No evidence of malignancy was identified, and ND was considered the most likely diagnosis.

**Figure 4 animals-15-02070-f004:**
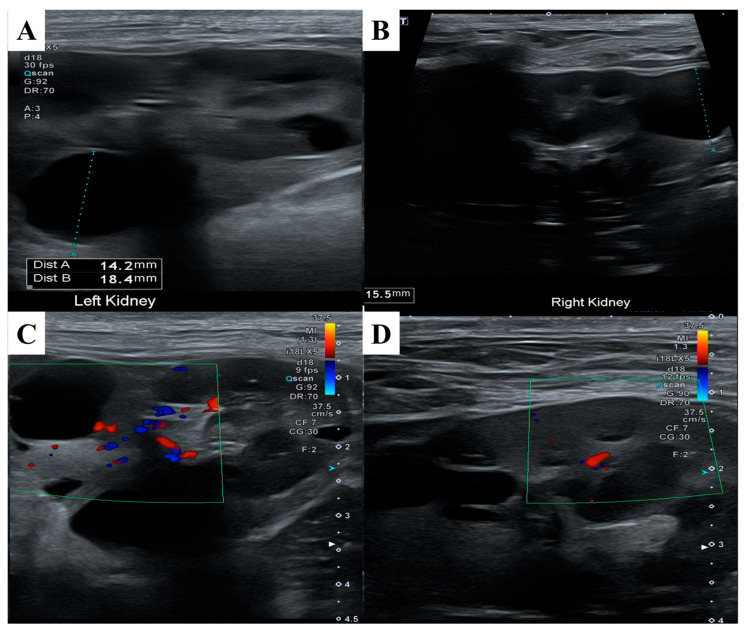
Abdominal ultrasonography findings of the kidneys. (**A**) Anechoic, thin-walled, well-marginated cystic structures were observed within and around the parenchyma of the left kidney. (**B**) Similar cystic structures were noted in the right kidney. (**C**) Color doppler imaging of the left kidney revealed no vascular flow within the cysts. (**D**) Color doppler imaging of the right kidney also showed no detectable vascular flow, supporting the benign nature of the cysts.

**Table 1 animals-15-02070-t001:** Overview of patient characteristics, FLCN mutation status, and presumed pathogenesis in reported cases.

Publication Year [Reference No.]	Breeds	Concomitance with Cysts or Masses	FLCN Gene Mutation	The Presumed Pathogenesis
1983 [1]	German Shepherd	Renal cystadenocarcinoma	Not tested	Autosomal dominant inheritance
1985 [2]	German Shepherd	Renal cystadenocarcinoma	Not tested	Genetic, likely an autosomal dominant disorder Cysts from tubular obstruction by epithelial proliferation
1985 [9]	German Shorthaired Pointer	None	Not tested	Congenital collagen hyperplasia (hamartoma) with excess fiber production
1986 [3]	German Shepherd	Renal cystadenocarcinoma	Not tested	Skin and kidney lesions likely share a genetic basis
1993 [11]	Golden Retriever	Renal cystadenoma	Not tested	Genetic factors or paraneoplastic syndrome Renal tumor-derived factors may trigger skin fibrosis
1997 [4]	German Shepherd	Renal cystadenocarcinoma	Not tested	Autosomal dominant tumor predisposition syndrome
1998 [10]	3 mixed-breed dogs and 1 Boxer	(1) Renal epithelial cysts with epithelial hyperplasia (2) Renal epithelial cysts (3) Renal cystadenomas (4) Renal cystadenocarcinoma and cystic adenomatous hyperplasia	Not tested	(1) Skin lesions as paraneoplastic effects of the renal tumor (2) Concurrent skin and kidney fibrosis from a shared genetic cause, with renal fibrosis causing tubular obstruction and cysts
2003 [5]	German Shepherd	Renal cystadenocarcinoma	Positive	H255R missense mutation in exon 7 of canine FLCN gene
2003 [6]	German Shepherd	Renal cystadenocarcinoma	Not tested	Primary pathogenesis: local TGF-β1 overexpression in skin
2008 [14]	Australian Cattle Dog	None	Not tested	Haploinsufficiency of FLCN gene
2013 [12]	Golden Retriever	None	Negative	Heterogeneous disorders from multiple genes or unknown causes
2019 [8]	Mixed breed	Renal cystadenocarcinoma	Positive	H255R heterozygous mutation in FLCN gene
2020 [15]	Mixed breed	Renal cystadenocarcinoma	Not tested	FLCN mutation with autosomal dominant inheritance
2020 [16]	Mixed breed	Renal cystadenoma	Not tested	Paraneoplastic syndrome or inherited disorder related to FLCN mutation
2022 [7]	German Shepherd	(1) Nodules in skin, intraperitoneal masses, lungs, bronchial lymph nodes, epiglottis, myocardium, kidney, adrenal glands, and intestinal wall (2) Cysts in intraperitoneal cavity	Not tested	(1) FLCN mutation causing autosomal dominant tumor suppressor loss (2) Local TGF-β1 overexpression (3) CD3 + T cells and CD163 + macrophages mediating inflammation and fibrosis
2024 [13]	Beagle	Renal cysts suspected; no biopsy performed	Negative	Other genetic factors or mechanisms beyond FLCN
Case at present	Bichon Frise	Renal cysts suspected; no biopsy performed	Negative	No FLCN mutation; variants found in BRCA2 genes

## Data Availability

This article contains all data generated or analyzed during this study. Further details or datasets are available from the corresponding author upon reasonable request.

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
