# Peer review of "Case Report: Atypical Nodular Dermatofibrosis and Renal Cysts in a Bichon Frise with a BRCA2 Mutation and No FLCN Mutation"

_animals, 2025, doi:10.3390/ani15142070_

Round 1
Reviewer 1 Report
Comments and Suggestions for Authors
Dear Authors, the manuscript provides a further contribution to the case history of this rare pathology without increasing current knowledge. For implementing knowledges it is useful provides immunotype collagen into its types Collagen I and Collagen IV for accepting the collagen origin in a dog with BRCA2 gene mutation instead the classical one related to FLCN gene.
There is an incongruence in the manuscript (see line 115-116 " ... serum biochemistry analysis ...but no significant abnormalities were identified." and see 287-288 "...accompained by elevated BUN and creatinine levels suggesting renal dysfunction."). Please, clarify the textual inconsistency.
Plesae, use an impersonal verbal form for increasing the scientific soundness of the manuscript (line 37, 62, 68).
Author Response
To clearly indicate the revisions made in response to Reviewer 1, the corresponding sections in the manuscript have been highlighted in yellow, using the same highlighter color as applied above the 'Reviewer 1' label in the response file.

Reviewer 2 Report
Comments and Suggestions for Authors
This is a very well written and comprehensive case report including a proposed novel pathogenesis of NDRC and an overview of the respective literature.
I recommend publication with only minor changes.
Abstract, line 26: I'd rather write: ... cytology of fine needle aspirates, histopathology of skin biopsies...
Introduction: Please check again the abbreviations. Sometimes you write NDF instead of ND.
Case description, line 78: what do you mean with cranial region?
line 90: please state the stain used for cytology
Figure 2: the very left arrowhead, which cell does it point to?
lines 111 and 115: You state the radiographic examination twice. Is this correct?
lines 130 ff (Histopathology): What about the inflammatory cells that were present in the cytology smears? Did you see any inflammatory cells in histopathology? If yes, please specify which, how many, where...
Figure 3: please only describe things that can be seen in the figures. lines 145-6: delete the text starting at ... expanding into the subcutis ... until ... dermis from the nodules. This cannot be distinguished in the small inset picture.
Please check the spelling, you use doppler and Doppler
Discussion, line 219: Please spell out BHD if used the first time
line 287: here you say BUN and creatinine levels are elevated, but before you state that they still are within reference range. So to me elevated is not the correct term. Maybe call it "rising"?
Conclusions, line 311: This finding does not "indicate" that RCND is not confined to a specific breed. There are previous reports stating this fact. But your finding supports this.
Author Response
To clearly indicate the revisions made in response to Reviewer 2, the corresponding sections in the manuscript have been highlighted in green, using the same highlighter color as applied above the 'Reviewer 2' label in the response file.

Reviewer 3 Report
Comments and Suggestions for Authors
The case report is original and very good.
The topic is very relevant, the aim of this research being to demonstrate that nodular dermatofibrosis (ND), a rare skin disease often linked to kidney cysts or tumors which typically affects specific breeds like the German Shepherd and is associated with mutations in the folliculin (FLCN) gene, was found in a Bichon Frise, a breed not previously linked to the disease and lacking the usual FLCN mutation.
Methodology is very good, authors using the modern tools of clinical imagistic, histology, blood biochemistry and genetic (molecular biology).
The results showed that no mutation in the traditionally implicated FLCN gene but multiple nonsynonymous mutations in the BRCA2 gene. This case suggests a potential association between BRCA2 gene mutations and the development of nodular dermatofibrosis.
Ultrasonography showed typical aspects of polycystic kidney disease.
The conclusions are supporting the research findings.
The references are relevant, but should be improved, to include more details about polycystic kidney disease in dog.
I suggest some corrections
In Introduction and Discussions authors should introduce some more details about polycystic kidney disease in dog.
Author Response
To clearly indicate the revisions made in response to Reviewer 3, the corresponding sections in the manuscript have been highlighted in Light blue, using the same highlighter color as applied above the 'Reviewer 3' label in the response file.

Round 2
Reviewer 1 Report
Comments and Suggestions for Authors
Dear Authors, the manuscript is ready for publication.